# A novel MARV glycoprotein-specific antibody with potentials of broad-spectrum neutralization to filovirus

Yuting Zhang[1,2†], Min Zhang[1†], Haiyan Wu[1†], Xinwei Wang[1,2†], Hang Zheng[1,2], Junjuan Feng[1,2], Jing Wang[1], Longlong Luo[1], He Xiao[1], Chunxia Qiao[1], Xinying Li[1], Yuanqiang Zheng[2], Weijin Huang[3], Youchun Wang[3], Yi Wang[4*‡], Yanchun Shi[2*‡], Jiannan Feng[1*‡], Guojiang Chen[1*‡]

[1]State Key Laboratory of Toxicology and Medical Countermeasures, Institute of Pharmacology and Toxicology, Beijing, China; [2]Inner Mongolia Key Lab of Molecular Biology, School of Basic Medical Sciences, Inner Mongolia Medical University, Hohhot, China; [3]Division of HIV/AIDS and Sex-transmitted Virus Vaccines, National Institutes for Food and Drug Control, Beijing, China; [4]Department of Hematology, Fifth Medical Center of Chinese PLA General Hospital, Beijing, China

*For correspondence:
yk091023@163.com (YW);
ycshi5388@163.com (YS);
fengjiannan1970@qq.com (JF);
jyk62033@163.com (GC)

†These authors contributed equally to this work
‡These authors also contributed equally to this work

Competing interest: The authors declare that no competing interests exist.

**Abstract** Marburg virus (MARV) is one of the filovirus species that cause deadly hemorrhagic fever in humans, with mortality rates up to 90%. Neutralizing antibodies represent ideal candidates to prevent or treat virus disease. However, no antibody has been approved for MARV treatment to date. In this study, we identified a novel human antibody named AF-03 that targeted MARV glycoprotein (GP). AF-03 possessed a high binding affinity to MARV GP and showed neutralizing and protective activities against the pseudotyped MARV in vitro and in vivo. Epitope identification, including molecular docking and experiment-based analysis of mutated species, revealed that AF-03 recognized the Niemann-Pick C1 (NPC1) binding domain within GP1. Interestingly, we found the neutralizing activity of AF-03 to pseudotyped Ebola viruses (EBOV, SUDV, and BDBV) harboring cleaved GP instead of full-length GP. Furthermore, NPC2-fused AF-03 exhibited neutralizing activity to several filovirus species and EBOV mutants via binding to CI-MPR. In conclusion, this work demonstrates that AF-03 represents a promising therapeutic cargo for filovirus-caused disease.

## eLife assessment

In this **valuable** study, the discovery and subsequent design of the AF03-NL chimeric antibody led to a tool for studying filoviruses and provides a possible blueprint for future therapeutics. In general, the data presented are **solid**, although further improvements can be made in the overall presentation of the results. The work will be of interest to virologists studying antibodies.

## Introduction

Filoviruses are nonsegmented negative-sense RNA viruses, comprised of six genera, Ebolavirus, Marburgvirus, Cuevavirus, Striavirus, Thamnovirus, and a recently discovered sixth genus, Dianlovirus (*Amarasinghe et al., 2019*; *Baseler et al., 2017*; *Kuhn et al., 2019*). The Marburgvirus genus consists of Marburg virus (MARV) and Ravn virus (RAVN) (*Amarasinghe et al., 2019*; *Bào et al., 2017*; *Ristanović et al., 2020*). The former includes three strains-- Uganda, Angola, and Musoke. The Ebola virus genus includes six distinct species including Zaire Ebola virus (EBOV), Bundibugyo virus (BDBV), Sudan virus (SUDV), Reston virus (RESTV), Taii Forest virus (TAFV) and Bombali virus (BOMV), the first

three of which cause severe hemorrhagic fevers (*Fan et al., 2020*; *Goldstein et al., 2018*). The genus Cuevavirus (Lloviu virus, LLOV) was isolated from Miniopterus schreibersii bats in Spain and Hungary and potently infected monkeys and human cells (*Negredo et al., 2011*; *Kemenesi et al., 2022*). The genus Měnglà virus (MLAV) was discovered in the liver of a bat from Mengla, Yunnan, China in 2019. So far, only an almost complete RNA sequence of the viral genome is available, there are no viable MLAVs isolated (*Yang et al., 2019*). MARV and EBOV infect humans and non-human primates, causing Marburg virus disease (MVD) and EBOV virus disease (EVD) with an incubation period of 2–21 days (*Mehedi et al., 2011*). The symptoms of MVD include severe headache and high fever rapidly within 5 days of the onset of symptoms, followed by diarrhea and vomiting, leading to up to 90% fatality rate (*Mehedi et al., 2011*). Therefore, MARV and EBOV have high potentials to cause a public health emergency.

Glycoprotein (GP) on the surface of filoviruses is a type I transmembrane protein and consists of GP1 and GP2 subunits (*Beniac and Booth, 2017*; *Lee and Saphire, 2009*). It is inserted into the virus envelope in the form of homotrimeric spikes (*Brauburger et al., 2012*) and is responsible for viral attachment and entry. The furin cleaves Marburg GP at the amino acid 435 into two subunits, GP1 and GP2, which remain linked by a disulfide bond (*Schafer et al., 2021*). GP1 contains a receptor binding domain (RBD), a glycan cap, and a heavily glycosylated mucin-like domain (MLD), which mediates binding to entry factors and receptors (*Hashiguchi et al., 2015*). GP2 has a partial MLD, a transmembrane domain for viral anchoring to the envelop surface, and a fusion peptide required for the fusion of virus and cell membranes (*Fusco et al., 2015*; *Gregory et al., 2011*; *Lee et al., 2017*). In the Ebola virus, the furin cleavage site is located at residue 501 and the entire MLD is attached to the GP1 subunit (*Volchkov et al., 2000*). Marburg virus contains 66 amino acids on GP2 that are absent from the Ebola virus MLD, and are called 'wings' due to their outward projection and flexibility (*Fusco et al., 2015*).

Currently, GP is a major target for antibodies validated in filovirus-infected animals and clinical trials because it is exposed on the surface of the virus and plays a key role in viral entry (*Dye et al., 2012*). Filoviruses initially enter cells by endocytosis or macropinocytosis (*Aleksandrowicz et al., 2011*; *Saeed et al., 2010*). Once inside the endosome, GP is cleaved by host cathepsins, and glycan cap and MLD are removed, enabling GP to bind to NPC1 (*Carette et al., 2011*; *Brecher et al., 2012*). Interestingly, Ebola viral entry requires cathepsin B cleavage (*Martinez et al., 2010*), which is redundant for MARV entry (*Gnirss et al., 2012*; *Misasi et al., 2012*). *Hashiguchi et al., 2015* proposed that the receptor binding domain was masked by glycan cap and MLD in the Ebola virus, whereas it was partially exposed in the Marburg virus.

To date, there is no licensed treatment or vaccine for Marburg infection, although a panel of antibodies with the potentials of neutralization has been isolated from a survivor subjected to MARV infection (*Flyak et al., 2015*). Herein, we utilized phage display technology to screen an antibody in a well-established antibody library (*Hu et al., 2021*; *Wang et al., 2022*) and obtained a novel human antibody with prominent neutralizing activity. Furthermore, NPC2 fusion at the N terminus of the light chain of this antibody potentiates broad-spectrum inhibition of cell entry of filovirus species and mutants.

## Results

### Characterization of AF-03

AF-03 was selected from a well-established phage surface display antibody library with immense diversity in the selected complementarity determining region (CDR) loops (*Liu et al., 2021*; *Duan et al., 2019*). We further subcloned VH and VL sequences of the antibody into a mammalian full-length immunoglobulin expression vector for full-length IgG expression. As shown in *Figure 1A*, AF-03 was eukaryotically expressed with the purity of over 95%. To determine the binding affinity, recombinant MARV GP without MLD was prepared (*Figure 1A*). ELISA analysis showed the remarkable binding of AF-03 to MARV GP (*Figure 1B*). Furthermore, SPR assay was done to determine the binding kinetics and showed that AF-03 bound to MARV GP with high affinity (AF-03 $K_D$ value was $4.71 \times 10^{-11}$ M, monovalent AF-03 Fab KD value was $2.15 \times 10^{-11}$ M) (*Figure 1C* and *Figure 1—figure supplement 1*). To identify determinants of MARV GP binding to AF-03, we utilized computer-guided homology modeling and molecular docking to generate computer models of MARV GP in a complex

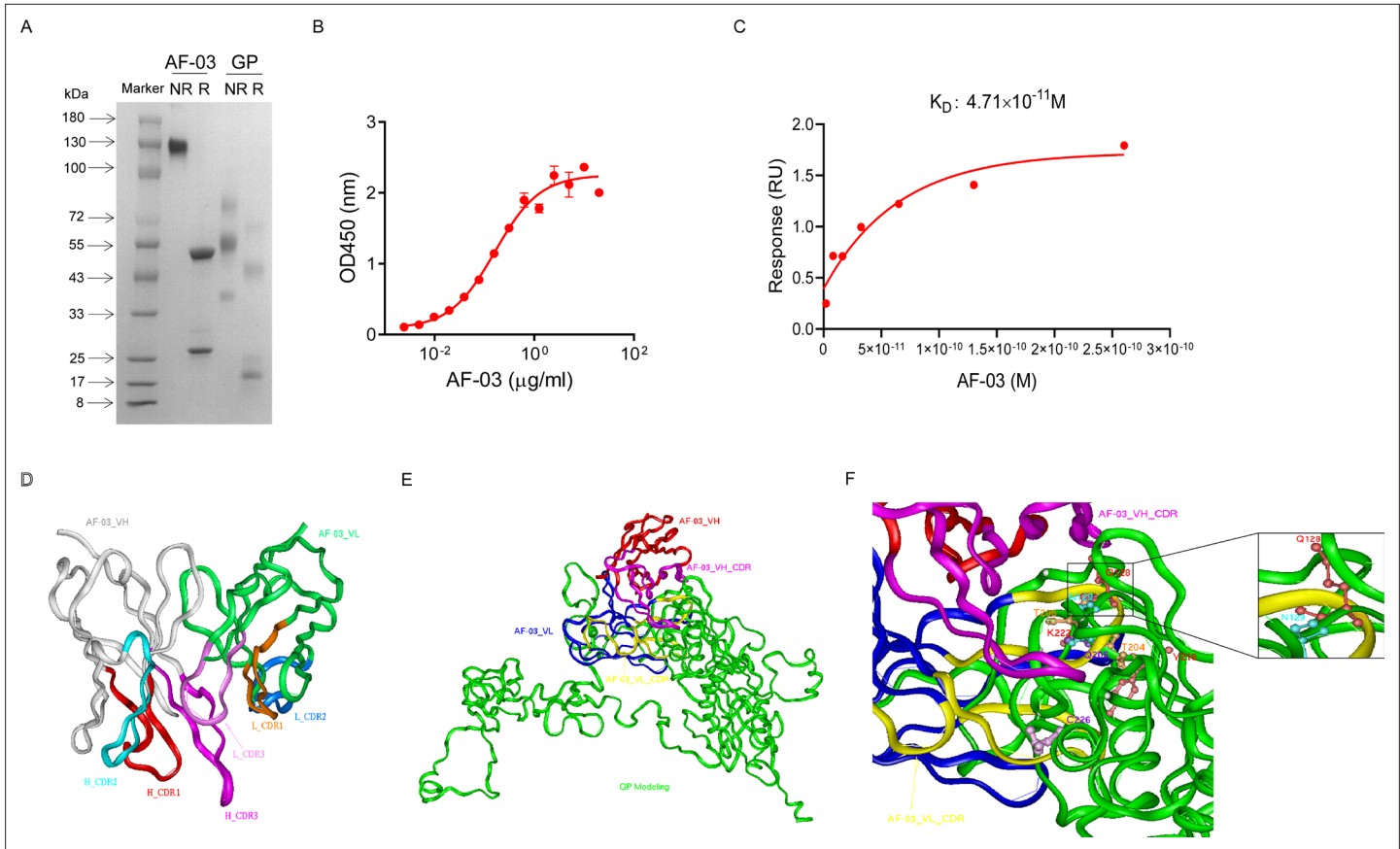

**Figure 1.** Binding activity of monoclonal antibody (mAb) AF-03 to Marburg virus (MARV) glycoprotein (GP) and its epitopes. (**A**) AF-03 and MARV GP proteins are examined by SDS-PAGE. NR, non-reducing; R, reducing. (**B**) The binding capacity of AF-03 to MARV GP is determined by ELISA. The absorbance is detected at 450 nm. (**C**) The binding kinetics of AF-03 to MARV GP is detected by SPR assay. Experiments are independently repeated at least three times, and the data from one representative experiment is shown. (**D**) The 3D ribbon structures of the AF-03 Fv fragment. The red ribbon denotes H-CDR1, the light blue denotes H-CDR2, the pink denotes H-CDR3, the orange denotes L-CDR1, the deep blue denotes L-CDR2, and the purple denotes L-CDR3. (**E**) AF-03 and MARV GP complex derived from theoretical modeling. The green ribbon denotes the orientation of the MARV GP fragment, the yellow denotes AF-03 VLCDR, the pink denotes AF-03 VHCDR, the deep blue denotes AF-03 VL and the red ribbon denotes AF-03 VH. (**F**) By molecular docking analysis of van der Waals interaction, intermolecular hydrogen bonding, polarity interaction, and electrostatic interaction, the key amino acid residues of MARV GP are screened. A zoom-in shows the predicted co-location of AF-03 CDR with Q128 and N129.

The online version of this article includes the following source data and figure supplement(s) for figure 1:

**Source data 1.** Raw image for *Figure 1A, D-F* and numerical data for *Figure 1B, C*.

**Figure supplement 1.** Binding activity of monovalent AF-03 Fab to Marburg virus (MARV) glycoprotein; (GP).

**Figure supplement 1—source data 1.** Raw image for *Figure 1—figure supplement 1A* and numerical data for *Figure 1—figure supplement 1B*.

with AF-03. Specifically, we obtained the theoretical 3D structure of AF-03 Fv (*Figure 1D*). Based on the 3D structure of AF-03 and MARV GP separately, the 3D complex structure of AF-03 and MARV GP achieved utilizing the molecular docking method (*Figure 1E*). Overall, these data suggest the potency of AF-03 binding to MARV GP.

## Epitope mapping of MARV GP bound to AF-03

Under CVFF forcefield, chosen steepest descent, and conjugate gradient minimization methods, after 30,000 steps of minimization with the convergence criterion 0.02 kCal/mol, the optimized structure of the AF-03 Fv was evaluated (*Figure 1D*). Using a Ramachandran plot, the assignment of the whole heavy atoms of the AF-03 Fv was in the credible range. Using the crystal complex structure of MR78 and GP protein as a model, based on the optimized theoretical 3D structure of MARV GP protein, the theoretical 3D complex structure of AF-03 and MARV GP protein was constructed (*Figure 1E*). Through analyzing the van der Waals interactions, inter-molecular hydrogen bonds, polar interactions, and

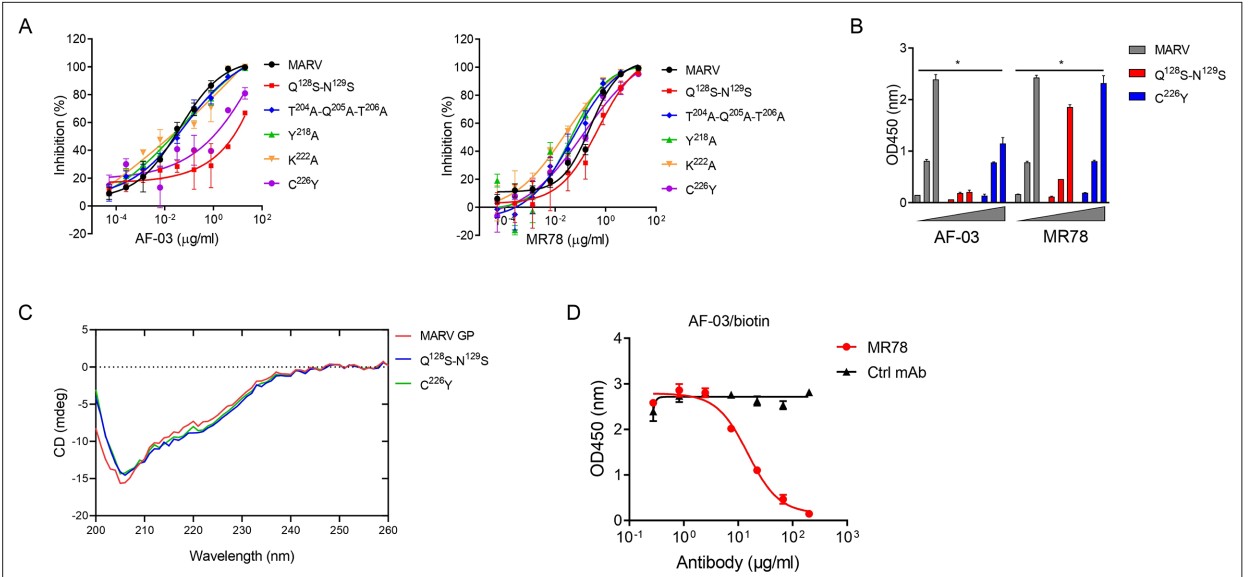

**Figure 2.** AF-03 Epitope Identification. (**A**) The neutralization activity of AF-03 or MR78 to mutated pseudovirus ($Q^{128}S$-$N^{129}S$, $Q^{204}A$-$T^{205}A$-$Q^{206}A$, $Y^{218}A$, $K^{222}A$, $C^{226}Y$) is evaluated in HEK293T cells. The inhibition rate is analyzed. (**B**) The binding of AF-03 and MR78 to mutant glycoprotein (GP) ($Q^{128}S$-$N^{129}S$ or $C^{226}Y$) is examined by ELISA, respectively. *$p<0.05$*. (**C**) Secondary structure of Marburg virus (MARV) GP and mutants are detected by circular dichroism (CD). (**D**) The epitope overlapping between AF-03 and MR78 is examined by the competitive ELISA. Experiments are independently repeated at least three times, and the data from one representative experiment is shown.

The online version of this article includes the following source data for figure 2:

**Source data 1.** Numerical data for *Figure 2A-D*.

electrostatic interactions between AF-03 and MARV GP, defining the binding region distance towards CDRs of AF-03 Fv fragment as 0.6 nm, the key amino acid residues of MARV GP were predicted for amino acid point mutations (*Figure 1F*). Based on the volume and character of amino acid residues, the residues $T^{204}$-$Q^{205}$-$T^{206}$, $Y^{218}$, and $K^{222}$ were mutated to alanine, and $Q^{128}$-$N^{129}$ were mutated to serine as well as $C^{226}$ was mutated to tyrosine. First, we investigated if this mutated MARV species was still sensitive to AF-03 treatment. The inhibition assay revealed the significant impairment of the neutralizing activity of AF-03 to MARV pseudovirus harboring $Q^{128}S$-$N^{129}S$ or $C^{226}Y$ compared with WT MARV and those loading other indicated mutations (*Figure 2A left panel*), which indicates that $Q^{128}$/$N^{129}$/$C^{226}$ functions as key amino acids responsible for AF-03 neutralization. Furthermore, we constructed the mutated MARV GP. ELISA assay showed that $Q^{128}S$-$N^{129}S$ or $C^{226}Y$ mutation significantly disrupted the binding of GP to AF-03, while the binding and neutralizing capacity of MR78 to mutant GP and pseudovirus harboring $C^{226}Y$ instead of $Q^{128}S$-$N^{129}S$, a mAb reported to be isolated from Marburg virus survivors (*Hashiguchi et al., 2015*), was comparable to WT counterparts (*Figure 2A right panel and B*). Furthermore, we analyzed the secondary structure of the MARV GP and its mutants. By circular dichroism, the structure of both mutants was not obviously different from that of the parental GP (*Figure 2C* and *Supplementary file 1*). Therefore, the weakened binding to the antibody was not due to the conformational change of GP caused by the mutation. Competitive ELISA showed that AF-03 and MR78 could compete with each other to bind to MARV GP (*Figure 2D*). Collectively, these results indicate the epitopes of these two mAbs overlapped partially.

## In vitro neutralizing activity of AF-03

Given the high binding affinity of AF-03 to MARV GP, we sought to determine whether AF-03 could impede pseudotyped MARV viral entry. An in vitro neutralization assay was developed based on a full-length MARV GP-pseudotyped virus using a HIV vector (pSG3.Δenv.cmvFluc). Liver and adrenal glands have been reported to be the early targets of MARV infection (*Brauburger et al., 2012*; *Shiflett and Marzi, 2019*). Therefore, we first tested the entry of MARV to hepatocyte cell line (Huh7) and renal cell line (HEK293T cells) by measuring the relative luciferase intensity. These two cell lines were susceptible to MARV cell entry. We used MR78 and cetuximab as positive and negative controls,

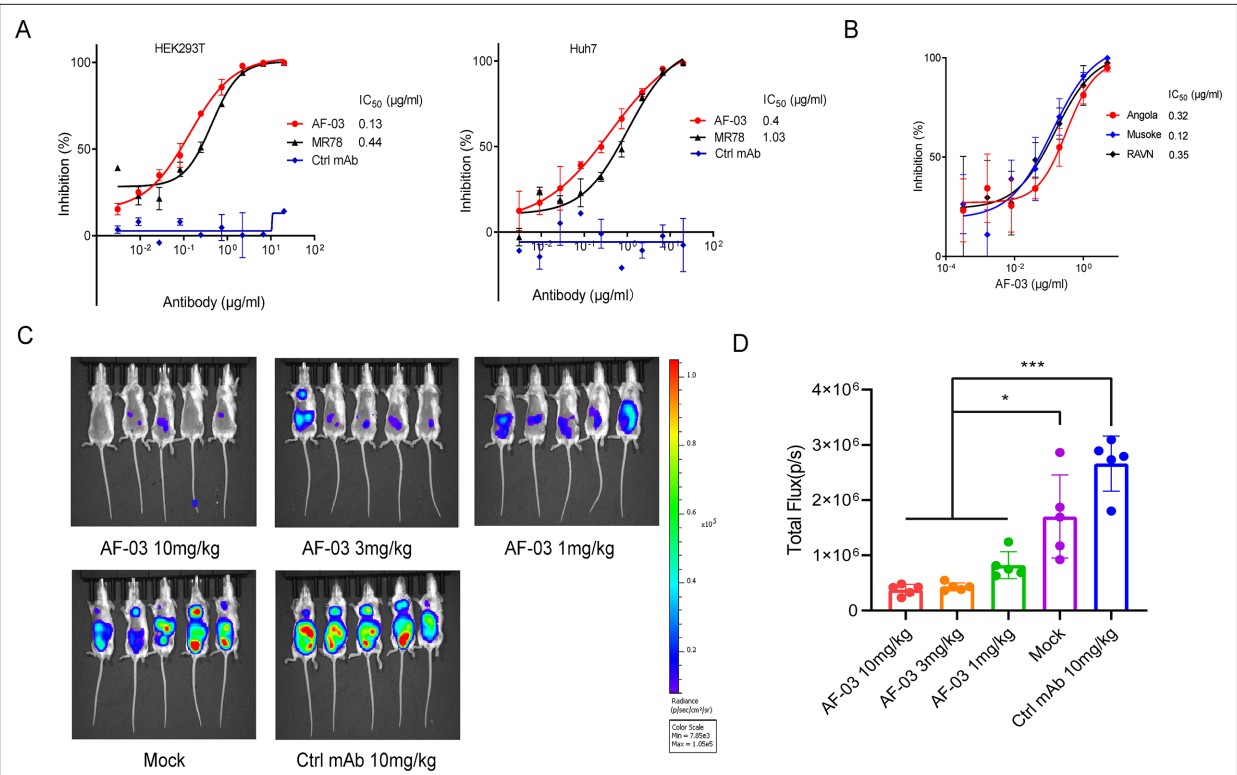

**Figure 3.** In vitro and in vivo neutralization of Marburg virus (MARV) pseudovirus infection by AF-03. (**A**) Pseudotypic MARV-Uganda is incubated with AF-03, MR78, or control mAb at 37 °C for 1 hr before infecting HEK293T cells (left) and hepatocyte cell line (Huh7) cells (right), respectively. Luciferase is assayed and inhibition rates are calculated. (**B**) Pseudotypic MARV-Angola, Musoke and Ravn virus (RAVN) infect HEK293T cells, respectively and neutralization activity of AF-03 to these species is determined. (**C**) AF-03 (10, 3, 1 mg/kg) is administrated at 24 and 4 hr before intraperitoneal injection of MARV pseudovirus. On day 4, bioluminescence signals are detected by an IVIS Lumina Series III imaging system. (**D**) The total radiance value is based on the luminescence of (**C**). *p<0.05, ***p<0.001. Experiments are independently repeated at least three times, and the data from one representative experiment is shown.

The online version of this article includes the following source data for figure 3:

**Source data 1.** Numerical data for *Figure 3A, B, D* and raw image for *Figure 3C*.

respectively. As expected, cetuximab had no effects on pseudotyped MARV entry. In contrast, AF-03 actively inhibited viral entry to HEK293T cells, with IC$_{50}$ value of 0.13 µg/ml. As well, IC$_{50}$ of MR78 was 0.44 µg/ml (*Figure 3A left panel*). In Huh7 cells, IC$_{50}$ of AF-03 and MR78 was 0.4 and 1.03 µg/ml, respectively (*Figure 3A right panel*). These results suggest that AF-03 has a stronger potency of neutralization than MR78. We also conducted AF-03 neutralization experiments on pseudotyped Angola, Musoke, and RAVN strains and showed a strong and comparable neutralizing ability to all these strains (IC$_{50}$ was 0.32, 0.12, and 0.15 µg/ml, respectively) (*Figure 3B*). Taken together, these data suggest that AF-03 harbors prominent neutralizing activity to MARV infection.

## In vivo preventive efficacy of AF-03

To verify the in vivo preventive efficacy, AF-03 was intravenously injected into mice before pseudotyped MARV exposure (–24 hr and –4 hr), respectively. The bioluminescence intensity was measured on day 4 after pseudovirus injection. As shown in *Figure 3C and D*, the AF-03-treated group displayed lower bioluminescence activity compared with the control group, while the treatment with control antibodies had no effects. Administration of 1 mg/kg AF-03 prior to the injection of MARV could decrease viral infection to approximately 50% level and increasing doses of AF-03 led to higher preventive efficacy. This indicates clearly that AF-03 is capable of preventing from MARV infection in a dose-dependent manner and represents a potential candidate for MARV treatment.

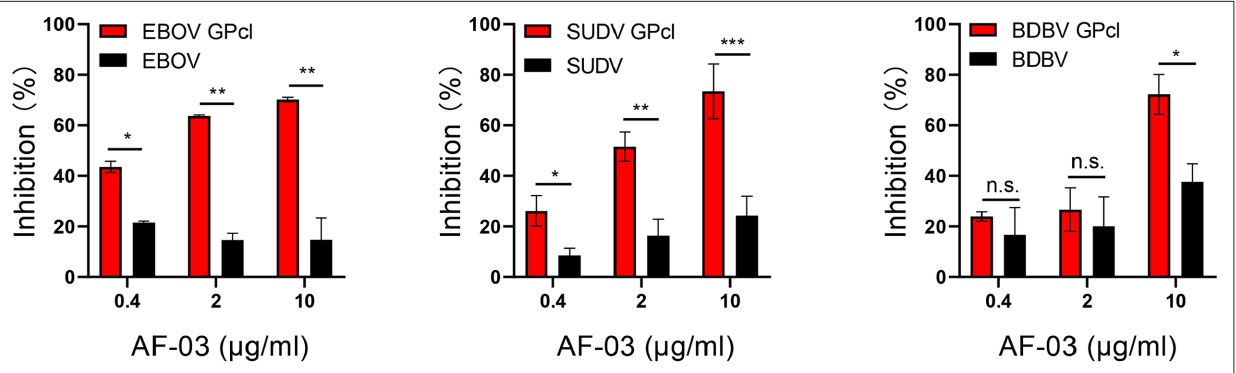

**Figure 4.** The neutralization activity of AF-03 to EBOV, SUDV, and BDBV harboring cleaved GP. Pseudotypic Ebola virus (EBOV), Sudan virus (SUDV), and Bundibugyo virus (BDBV) are processed with thermolysin at 37 °C. Inhibition of these ebola virus entry harboring glycoprotein (GP) or GPcl by AF-03 is examined by luciferase assay. *p<0.05, **p<0.01, ***p<0.001. Experiments are independently repeated at least three times, and the data from one representative experiment is shown.

The online version of this article includes the following source data and figure supplement(s) for figure 4:

Source data 1. Numerical data for *Figure 4*.

Figure supplement 1. Comparison of the cellular entry capacity of ebolavirus harboring cleaved or intact glycoprotein (GP).

Figure supplement 1—source data 1. Numerical data for *Figure 4—figure supplement 1*.

## AF-03 impedes cell entry of EBOV, SUDV, and BDBV harboring GPcl

Given the close structural similarity of the Marburg virus to the ebola virus, to determine whether AF-03 was also available to the treatment of EBOV infection, we conducted neutralization of pseudo-typed EBOV, SUDV, and BDBV by AF-03. In addition, considering that glycan cap or mucin-like domain are known to mask the putative receptor-binding domain on EBOV, SUDV, and BDBV GP and thus impede the engagement between AF-03 and GP (*Hashiguchi et al., 2015*), glycan cap and mucin-like domain were enzymatically cleaved by digesting GP with 250 µg/ml thermolysin at 37 °C. The infection of pseudotyped ebolavirus harboring cleaved GP to host cells was comparable or stronger than those containing intact GP (*Figure 4—figure supplement 1*). Intriguingly, the inhibitory function of AF-03 on cell entry of all three species of ebolavirus bearing cleaved GP was much stronger than those bearing uncleaved GP (*Figure 4*), which suggests that AF-03 has therapeutic potentials for EBOV, SUDV, and BDBV infection.

## The potency of NPC2-fused AF-03 to be delivered into the endosome

Given the inability of AF-03 to transport into the endosomal compartment where intact GP is cleaved by cathepsin B/L, we engaged NPC2 to the N-terminus of the light chain of AF-03 (*Figure 5—figure supplement 1A*), according to a protocol described previously (*Wirchnianski et al., 2021*). As well, we produced the 1–3 domain of CI-MPR (*Figure 5—figure supplement 1A*), which is a ligand for NPC2 and expressed on the cellular and endosomal membrane (*Bohnsack et al., 2009*). The results showed that NPC2-fused AF-03 (termed AF03-NL), rather than AF-03, bound to CI-MPR1-3 (*Figure 5—figure supplement 1B*). Next, we investigated the internalization of AF03-NL. AF03-NL or AF-03 was incubated with HEK293T cells, which expressed CI-MPR (*Figure 5—figure supplement 2A*), at 4°C for attachment. As expected, AF03-NL instead of AF-03 adhered to the cell surface, detected by fluorescence-labelled secondary IgG. Upon endocytosis, the fluorescence on the cell surface decreased dramatically, implying the occurrence of AF03-NL internalization (*Figure 5A*). To further address this issue, AF-03 and AF03-NL were labelled by antibody internalization reagent and pHrodo Red dye, respectively that is sensitive to acidic niche. Consequently, APC and pHrodo Red-conjugated AF03-NL was observed in the acidic endosomal compartment by flow cytometry and fluorescence microscopy, respectively (*Figure 5B and C*). In contrast, AF-03 was not seen in the endosome.

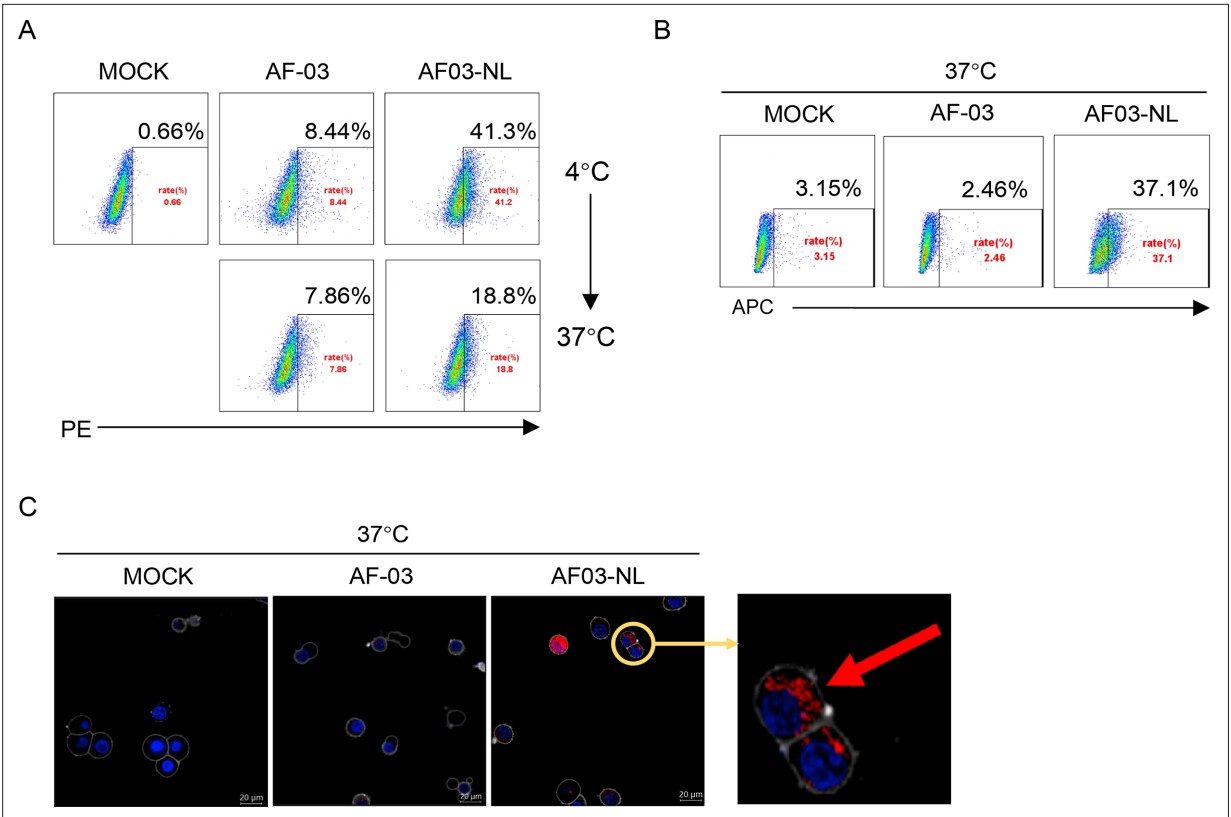

**Figure 5.** Cellular internalization of AF03-NL. AF-03, AF03-NL, or human IgG1 isotype (MOCK) is incubated with cells at 4℃ for 1 hr to prevent internalization and then at 37℃ for another 2 hr to allow internalization. PE-conjugated secondary antibody is added prior to analysis by flow cytometry. (**B,C**) Antibody internalization reagent and pHrodo Red-labeled AF-03 or AF03-NL is incubated with cells at 37℃ for 1 hr and analyzed by flow cytometry (B) and fluorescence microscopy, (**C**) respectively. The red arrow denotes internalized AF03-NL. Experiments are independently repeated at least three times, and the data from one representative experiment is shown.

The online version of this article includes the following source data and figure supplement(s) for figure 5:

**Source data 1.** Raw image for *Figure 5A-C*.

**Figure supplement 1.** Characterization of AF03-NL and CI-MPR1-3.

**Figure supplement 1—source data 1.** Raw image for *Figure 5—figure supplement 1A* and numerical data for *Figure 5—figure supplement 1B*.

**Figure supplement 2.** CI-MPR expression in HEK293T and hepatocyte cell line (Huh7) cells.

**Figure supplement 2—source data 1.** Raw image for *Figure 5—figure supplement 2A, B*.

## Pan-filovirus inhibition of cell entry by AF03-NL via engagement between NPC2 and CI-MPR

First, we compared the binding of AF03-NL/AF-03 to MARV GP. ELISA showed relatively weak binding activity of AF03-NL compared with AF-03 (*Figure 6—figure supplement 1A*). We, thereafter, evaluated the neutralizing activity of AF03-NL/AF-03 to MARV pseudovirus. Intriguingly, AF03-NL showed stronger neutralizing activity than AF-03 (The IC$_{50}$ was 0.057 and 0.284 μg/ml, respectively) (*Figure 6—figure supplement 1B*), which may be attributed to sustained tethering of AF03-NL to pseudovirus at extracellular space as well as endosomal compartment. Second, we compared the neutralizing activity of AF03-NL and AF-03 to a series of filovirus species. AF03-NL displayed superior neutralizing activity to the other nine filovirus species. While, no or weak inhibition of entry by AF-03 was found (*Figure 6A*). Furthermore, AF03-NL, instead of AF-03, also actively inhibited cell entry of 17 EBOV mutants that were detected previously in natural hosts (*Figure 6B*).

To investigate the mechanisms underlying the potency of AF03-NL, we produced NPC2 protein (*Figure 7A*) and then examined the inhibition of EBOV entry by AF03-NL or the mixture of AF-03 and NPC2. AF-03, NPC2 alone or in combination did not inhibit EBOV entry. Conversely, AF03-NL actively impeded this process (*Figure 7B*). To clarify whether this effect is CI-MPR-dependent, CI-MPR in

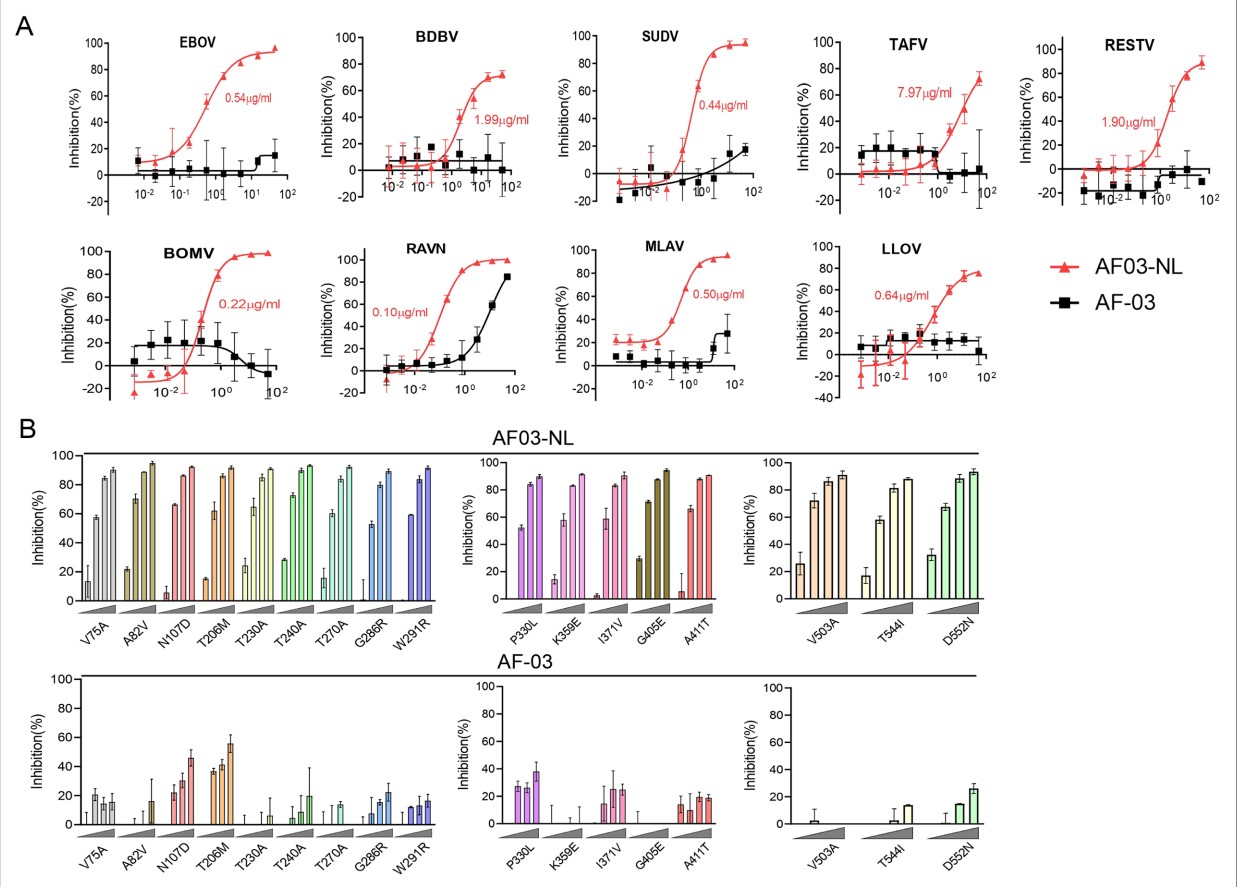

**Figure 6.** Pan-filovirus entry inhibition by AF03-NL. (**A,B**) AF-03 or AF03-NL (50–0.0007 μg/ml, fourfold dilution) is incubated with HEK293T cells at 37 °C for 2 hr prior to exposure to pseudotypic filovirus species (**A**) and Ebola virus (EBOV) mutants (**B**). Luciferase is assayed and inhibition rates are calculated. Experiments are independently repeated at least three times, and the data from one representative experiment is shown.

The online version of this article includes the following source data and figure supplement(s) for figure 6:

**Source data 1.** Numerical data for *Figure 6A, C*.

**Figure supplement 1.** Comparable binding and inhibitory activity of AF03-NL and AF-03.

**Figure supplement 1—source data 1.** Numerical data for *Figure 6—figure supplement 1A, B*.

HEK293T cells was silenced (*Figure 7C*). The result showed that CI-MPR knockdown rendered significant abrogation of the inhibitory ability of AF03-NL (*Figure 7C*). We also introduced CI-MPR into the Huh7 cell line that is null for this receptor (*Figure 5—figure supplement 2B*). The inhibitory effects of AF03-NL were augmented in CI-MPR-overexpressed cells compared with empty vector-introduced counterparts (*Figure 7D*). Taken together, these data indicate that the inhibitory potency of AF03-NL is dependent on the interaction between NPC2 and CI-MPR.

## Discussion

The Marburg virus was initially identified after simultaneous outbreaks in Marburg and Frankfurt in Germany in 1967 (*Hashiguchi et al., 2015*; *Abir et al., 2022*). To date, there have been a dozen outbreaks of Marburg virus infection in humans (*Araf et al., 2023*). Giving the recurrence of Marburg virus outbreaks and its high virulence and lethality, there is an urgent need to develop prophylactic and therapeutic interventions for Marburg infections. MARV GP is a surface viral protein, which is responsible for host receptor binding and cell entry thus provides an attractive target for the development of antagonists. *Flyak et al., 2015* screened several MARV GP-specific neutralizing antibodies from the PBMC samples of a MARV-infected survivor, which achieved 100% protection in mice subjected to mouse-adapted MARV challenge. The MARV GP-specific antibody cocktail was also developed, three

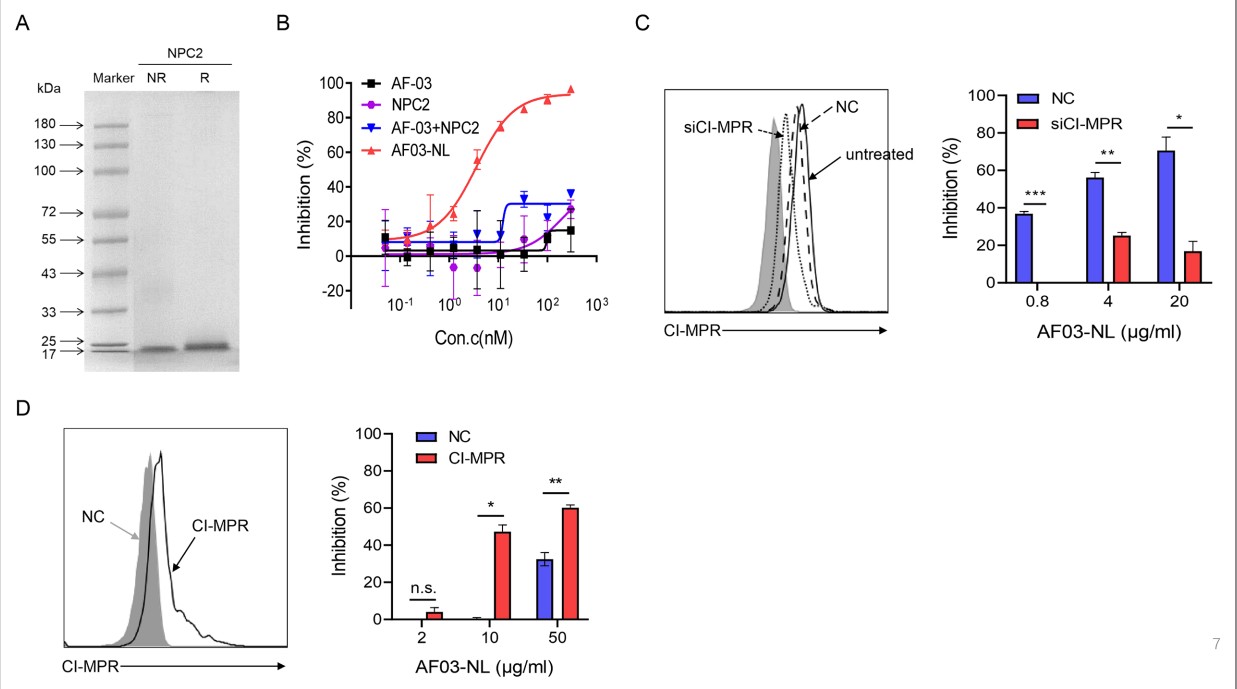

**Figure 7.** The requirement of CI-MPR for the neutralization activity of AF03-NL. (**A**) Niemann-Pick C2 (NPC2) protein is examined by SDS-PAGE. NR, non-reducing; R, reducing. (**B**) AF03-NL, AF-03, NPC2 alone, or equimolar combination of AF-03 and NPC2 is incubated with HEK293T cells at 37 °C for 2 hr prior to exposure to pseudotypic Ebola virus (EBOV). Luciferase is assayed and inhibition rates are calculated. (**C**) HEK293T cells are treated with siRNA-CI-MPR or negative control vector (NC), respectively and CI-MPR expression is detected by flow cytometry. AF03-NL is incubated with siCI-MPR or NC-treated HEK293T cells at 37 °C for 2 hr, respectively prior to exposure to pseudotypic EBOV. (**D**) CI-MPR is introduced into hepatocyte cell line (Huh7) cells and its expression is detected by flow cytometry. AF03-NL is incubated with CI-MPR or NC-knockin Huh7 cells at 37 °C for 2 hr, respectively prior to exposure to pseudotypic EBOV. Luciferase is assayed and inhibition rates are calculated. Experiments are independently repeated at least three times, and the data from one representative experiment is shown.

The online version of this article includes the following source data for figure 7:

**Source data 1.** Raw image for *Figure 7A, C* left panel, *Figure 7D* left panel and numerical data for *Figure 7B, C* right panel, *Figure 7D* right panel.

mAbs cocktail could protect hamsters from lethal hamster-adapted MARV infection, while treatment with either one or two antibodies failed (*Marzi et al., 2018*).

In this study, we selected an antibody from a human antibody phage library and the affinity constant reached the $10^{-11}$M level. The neutralizing activities of the antibody were demonstrated by utilizing the pseudotyped MARV Uganda strain. The results showed that AF-03 effectively inhibited HIV vector (pSG3.Δenv.cmvFluc) pseudotyped MARV viral entry at $IC_{50}$ of 0.13 and 0.4 µg/ml in HEK293T and Huh7, respectively. Furthermore, compared with the control antibody, AF-03 exhibited a protective property against pseudovirus infection in mice. Epitope mapping results showed that $Q^{128}$-$N^{129}$ and $C^{226}$ of GP was the binding and functional epitopes that interacted with AF-03, which means AF-03 targeting the interface of GP-NPC1 interaction, considering that $N^{129}$ is known to be located in the NPC1 binding domain. RBD is highly conserved among filovirus species, so it is an attractive target for broadly effective anti-filovirus drug development (*Densumite et al., 2021*). We found that AF-03 also bound to EBOV GPcl and could neutralize ebola viruses bearing cleaved GP in vitro, suggesting that AF-03 represents a good candidate for endosome-delivering strategy by ligation to another mAb against a surface-exposed EBOV GP epitope or a ligand peptide for host cation-independent mannose-6-phosphate receptor (*Wirchnianski et al., 2021*), which will ultimately afford cross-reactivity against multiple filovirus species. Accordingly, we designed NPC2-fused AF-03 and demonstrated its broad-spectrum inhibitory capacity to filovirus species and EBOV mutants. Future investigations on the inhibition of AF03-NL to authentic virus infection in vitro and in vivo are warranted.

Overall, our study identified a high-affinity anti-MARV antibody AF-03 targeting a conserved and hidden site at the filovirus GPcl-NPC1 interface, which was capable of neutralizing MARV infection both in vitro and in vivo. Furthermore, AF-03 may be a potential candidate for the effective protection

against pan-filovirus species infection. Investigations on AF-03 treatment of mice challenged by authentic virus are undergoing.

# Materials and methods

**Key resources table**

| Reagent type (species) or resource | Designation | Source or reference | Identifiers | Additional information |
|---|---|---|---|---|
| Cell line (*H. sapiens*) | HEK293T | ATCC | RRID:CVCL_0063 | |
| Cell line (*H. sapiens*) | Huh7 | ATCC | RRID:CVCL_U443 | |
| Cell line (Chinese hamster) | ExpiCHO-S | Thermo | RRID:CVCL_5J31 | |
| Gene (Marburg virus) | Uganda | This paper | GenBank: AFV31370.1 | |
| Gene (Marburg virus) | Angola | This paper | Uniprot: Q1PD50 | |
| Gene (Marburg virus) | Musoke | This paper | Uniprot: P35253 | |
| Gene (Marburg virus) | RAVN | This paper | Uniprot: Q1PDC7 | |
| Gene (Ebola virus) | TAFV | This paper | Uniprot: Q66810 | |
| Gene (Ebola virus) | RESTV | This paper | Uniprot: Q66799 | |
| Gene (Ebola virus) | BOMV | This paper | GenBank: YP_009513277.1 | |
| Gene (Ebola virus) | EBOV | Kindly gifted by the China Institute for Food and Drug Control | GenBank: AHX24649.2 | |
| Gene (Ebola virus) | BDBV | Kindly gifted by the China Institute for Food and Drug Control | GenBank: YP_003815435 | |
| Gene (Ebola virus) | SUDV | Kindly gifted by the China Institute for Food and Drug Control | GenBank: YP_138523.1 | |
| Gene (Cueva virus) | LLOV | This paper | GenBank: JF828358.1 | |
| Gene (Dianlo virus) | MLAV | This paper | GenBank: YP_010087186.1 | |
| Gene | pSG3. Δenv. cmvFluc | Kindly gifted by the China Institute for Food and Drug Control doi: 10.1038/srep45552 | | |
| Gene | pFRT-KIgG1 | Thermo | Cat# V601020 | |
| Commercial assay or kit | Dulbecco's modified Eagle's medium (DMEM) | Gibco | Cat# 11965e092 | |
| Commercial assay or kit | Pen Strep | Gibco | Cat# 15140 | |
| Commercial assay or kit | Fetal bovine serum | Gibco | Cat# 10099 | |
| Commercial assay or kit | ExpiCHO Expression Medium | Gibco | Cat# A29100 | |
| Commercial assay or kit | ExpiFectamine CHO Transfection Kit | Gibco | Cat# A29129 | |
| Commercial assay or kit | Nickel column | Cytiva | Cat# 11003399 | |
| Commercial assay or kit | Soluble TMB Kit | CWBIO | Cat# CW0050S | |
| Commercial assay or kit | Transfection reagent | JetPRIME | Cat# 25Y1801N5 | |
| Commercial assay or kit | Bright-Glo luciferase reagent | Promega | Cat# E6120 | |
| Commercial assay or kit | thermolysin | Sigma | Cat# T7902 | |
| Commercial assay or kit | Phosphoramidon | Sigma | Cat# R7385 | |
| Commercial assay or kit | D-luciferin | PerkinElmer | Cat# 122799 | |

*Continued on next page*

*Continued*

| Reagent type (species) or resource | Designation | Source or reference | Identifiers | Additional information |
|---|---|---|---|---|
| Commercial assay or kit | iQue Human Antibody Internalization Reagent | Sartorius | Cat# 90564 | |
| Commercial assay or kit | pH-sensitive pHrodo red succinimidyl ester | Thermo | Cat# P36600 | |
| Commercial assay or kit | polylysine | Beyotime | Cat# ST508 | |
| Commercial assay or kit | DiD | Thermo | Cat# V22887 | |
| Commercial assay or kit | Hoechst33342 | Thermo | Cat# H1398 | |
| Commercial assay or kit | Pierce Fab Preparation Kit | Thermo | Cat# 44985 | |
| Antibody | horseradish peroxidase (HRP)-labeled goat anti-human IgG secondary antibody | Invitrogen | Cat# A18817 RRID:AB_1640167 | Elisa: 1:6000 |
| Antibody | Horseradish peroxidase (HRP)-labeled Streptavidin | Thermo | Cat# S911 RRID:AB_795453 | Elisa: 1:10000 |
| Antibody | PE-labeled anti-human IgG Fc secondary antibody | Biolegend | Clone M1310G05 Cat# 41070 | |
| Antibody | FITC-conjugated anti-CI-MPR antibody | Biolegend | Clone QA19A18 Cat# 364207 | |
| Experimental animals | BALB/c | Beijing Vital River Laboratory Animal Technology | | Four-week-old, female |

## Cell lines and plasmids

Human embryonic kidney cells HEK293T and human hepatoma cells Huh7 were purchased from ATCC. These cell lines were cultured in Dulbecco's modified Eagle's medium (DMEM) (Gibco) supplemented with 100 units/ml penicillin, 100 units/ml streptomycin (Gibco), and 10% fetal bovine serum (Gibco) in a humidified atmosphere (5% $CO_2$, 95% air) at 37 °C. ExpiCHO-S cells were purchased from Gibco and cultured in ExpiCHO Expression Medium (Gibco) in a humidified atmosphere (8% $CO_2$, 92% air) on an orbital shaker platform. The cells were authenticated using short tandem repeat (STR) profiling and were also tested for mycoplasma contamination.

MARV, Angola, Musoke, RAVN, TAFV, RESTV, BOMV, MLAV, LLOV GP plasmids were synthesized by GENEWIZ and then cloned into the expression vector pcDNA3.1. EBOV, BDBV, SUDV GP, and HIV-based vector pSG3. Δenv. cmvFluc plasmids were kindly gifted by the China Institute for Food and Drug Control.

## Preparation of full-length antibody and antigen

AF-03 was selected from a human phage antibody library, which displays on the surface of M13 bacteriophage particles. Screening procedures were described in detail previously (*Liu et al., 2021*; *Duan et al., 2019*). Phage antibodies that bound to MARV GP protein were obtained to express full-length IgG using a standard protocol. In brief, the VH and VL regions of AF-03 were constructed into a mammalian full-length immunoglobulin expression vector pFRT-KIgG1 (Thermo) to generate plasmid AF-03. The human NPC2 gene (aa20-151) was linked to the VL of AF-03 by a short linker 'TVAAP' and then constructed into pFRT-KIgG1 (designated as AF03-NL). The AF-03 and AF03-NL plasmid were transfected into ExpiCHO-S cells using the ExpiFectamine CHO Transfection Kit (Gibco) following the manufacturer's instructions. Purification was performed using the ÄKTA prime plus system (GE Healthcare) with protein A column (GE Healthcare). Fab generation from IgG using the Pierce Fab Preparation Kit (Thermo) following the manufacturer's instructions. MARV GP (Uganda strain) (aa 20–648, Δ277–455), CI-MPR1-3 (aa36-466), and NPC2 (aa20-151) gene with six histidine-tagged at C-terminus was cloned into mammalian expression vector pcDNA3.1, respectively and then transfected into HEK293T cells. MARV GP was purified using a nickel column (Cytiva). The concentration of proteins and antibodies were quantified by bicinchoninic acid (BCA) method.

## ELISA

The 96-well plates were coated with 2 μg/ml MARV-GP and mutated MARV GP ($Q^{128}S$-$N^{129}S$/$C^{226}Y$), respectively and incubated overnight at 4 °C. Wells were washed for three times and blocked for 1 hr at 37 °C. A series of 12 concentrations of AF-03 and MR78 (starting at 20 μg/ml, twofold dilution) were added and incubated for 1 hr at 37 °C. Bound antibodies were detected with horseradish peroxidase (HRP)-labeled goat anti-human IgG secondary antibody (Invitrogen) at 37 °C for 30 min. Binding signals were visualized using a TMB substrate (CWBIO) and the reaction was stopped by adding 2 N $H_2SO_4$. The light absorbance at 450 nm was measured by a microplate reader (Thermo Fisher Scientific).

For competitive ELISA, biotinylated AF-03 (1 μg/ml) was coated. MR78 and control mAb (Herceptin) in threefold serial dilution (ranging from 200 to 0.27 μg/ml) and added to the plates. After 1 h incubation at 37 °C, the plates were washed and the bound biotin-AF-03 was detected by adding horseradish peroxidase (HRP)-labeled Streptavidin (Thermo). After a further 30 min incubation at 37 °C, the plates were washed and TMB was added. The reaction was stopped by adding 2 N $H_2SO_4$. Absorbance was measured at 450 nm using a plate reader.

## SPR analysis of antibody affinity

The SPR (surface plasmon resonance) analysis was performed using a Biacore T200 machine with CM5 chips (GE Healthcare) at room temperature (25 °C). All the proteins used in SPR analysis were exchanged to BIAcore buffer, consisting of PBS-P+, with 0.5% Surfactant P20 and 0.5% DMSO, pH 7.4. The chip was subsequently immobilized with MARV GP in sodium acetate, pH 5.0, and then blocked with 1 M ethanolamine, pH 8.0. AF-03 were diluted by running a buffer ranging from 0.26 to 0.002 nM. The chip was regenerated with glycine- HCl (pH 2.0, 10 mM). Data were analyzed with Biacore T200 Evaluation Software.

## Computer-guided homology modeling and molecular docking

To uncover the potential epitope of the GP protein, the binding mode between AF-03 and MARV GP protein was analyzed theoretically as follows: The three-dimensional theoretical structure of fragment variable (FV) was constructed using computer-guided homology modeling approach (Insight II 2000 software, MSI Co., stored in IBM workstation) based on the amino acid sequences of the variable structural domains of the heavy and light chains of AF-03, and the conserved regions (the framework region of the antibody variable domain) and loop structural domains (the CDR region of the antibody variable domain) were identified. The 3D structure of the AF-03 Fv fragment was optimized under the consistent valence force field (CVFF) using the steepest descent and conjugate gradient minimization methods. The final minimized 3D structure was evaluated by means of Ramachandran diagrams. In addition, the 3D theoretical structure of the MARV GP protein was obtained using Alphafold 2 software online and optimized using the CVFF force field in Insight II 2000 software. Under the molecular docking method, using the crystal complex structure of MR78 and GP protein (PDB code: 5uqy) as model (*Hashiguchi et al., 2015*), the 3D complex structures AF-03 Fv fragment and GP were obtained and optimized. With the determined 3D structure of the AF-03 Fv fragment and GP, 50-ns molecular dynamics were performed with the Discovery_3 module. All calculations were performed using Insight II 2000 software (MSI Co., San Diego) with IBM workstation.

## Pseudovirus preparation

HIV vector (pSG3.Δenv.cmvFluc) bearing MARV, mutated MARV ($Q^{128}S$-$N^{129}S$, $T^{204}A$-$Q^{205}A$-$T^{206}A$, $Y^{218}A$, $K^{222}A$, and $C^{226}Y$), EBOV (parental and 17 mutants indicated), SUDV, BDBV, TAFV, RESTV, BOMV, RAVN, MLAV, and LLOV GPs were prepared by liposome-mediated transfection of HEK293T cells using transfection reagent (JetPRIME), respectively. Cells were seeded in six-well plates at a density of $7 \times 10^5$ cells/well and transfected with 2 μg plasmids (0.4 μg GP and 1.6 μg HIV vector) when cells reached 60–80% confluence. Supernatants were collected 48 hr after transfection, centrifuged to remove cell debris at 3000 rpm for 10 min, filtered through a 0.45 μm-pore filter (Millipore, SLHUR33RB), and stored at –80 °C.

## Pseudovirus entry and antibody neutralization assay

Huh7 and HEK293T cells ($3 \times 10^4$ cells/100 µl/well) were infected with 100 µl pseudovirus, which contained a luciferase reporter gene, respectively. The luciferase activity was measured in a fluorescence microplate reader (Promega). The operation steps were following: after 36 hr incubation at 37 °C, 100 µl of culture medium were discarded and added with 100 µl of Bright-Glo luciferase reagent (Promega) in each well. Mixtures were transferred to 96-well whiteboards after a 2 min reaction to detect the relative luciferase intensity.

For AF-03 neutralization of MARV assays, 50 µl mAb (starting at 20 µg/ml) was threefold serially diluted and separately mixed with MARV pseudovirus at the same volume. The mixture was incubated at 37 °C for 1 hr, followed by the addition of 100 µl cells ($3 \times 10^4$ cells/well). 50% of maximal inhibitory concentration was defined as $IC_{50}$. $IC_{50}$ values were determined by non-linear regression with least-squares fit in GraphPad Prism 8 (GraphPad Software).

In terms of AF-03 neutralization of ebola virus assays, pseudovirus (EBOV, SUDV, and BDBV) were processed with thermolysin as previously described (*Markosyan et al., 2016*). Briefly, Pseudotyped ebola virus were incubated at 37 °C for 1 hr with 200 µg/ml thermolysin (Sigma). The reaction was stopped by the addition of 400 µM Phosphoramidon (Sigma) on ice for 20 min. The remaining steps followed AF-03 neutralization of MARV assays.

For AF03-NL neutralization assays, 50 µl/well of serial diluted AF03-NL and AF-03 (starting at 50 µg/ml) were incubated with cells at 37 °C for 2 hr to enable internalization of the antibodies. 50 µl/well of diluted pseudovirus was then added to a 96-well plate and incubated at 37 °C for 36 hr. Bright-Glo luciferase reagent was added to detect the relative luciferase intensity.

## Bioluminescent imaging in vivo

Four-week-old female BALB/c mice were purchased from Beijing Vital River Laboratory Animal Technology Co. Ltd. The animal study was reviewed and approved by the Institutional Animal Care and Use Committee of the Academy of Military Medical Sciences (IACUC-DWZX-2020–697). Mice were intraperitoneally injected with MARV pseudovirus (0.2 ml/mouse). AF-03 (10, 3, or 1 mg/kg) or control antibody Herceptin (10 mg/kg) were injected via intravenous route 24 hr and 4 hr before the pseudovirus injection, respectively. Bioluminescent signals were monitored at Day 4. Briefly, D-luciferin (150 mg/kg body weight) (PerkinElmer) was intraperitoneally injected into the mice, and then exposed to Isoflurane alkyl for anesthesia. Bioluminescence was measured by the IVIS Lumina Series III Imaging System (Xenogen, Baltimore, MD, USA) with the living Image software. The signals emitted from different regions of the body were measured and presented as average fluxes. All data are presented as mean values ± SEM.

## Evaluation of internalization

HEK293T cells ($3 \times 10^5$ cells/well) were washed twice with cold PBS and the supernatant was discarded. AF-03/AF03-NL/human IgG1 isotype (20 µg/ml) was added and incubated at 4 °C for 30 min. One group was transferred to 37 °C for internalization for 30 min, while the other group continued to be incubated at 4 °C for 30 min to adhere to the cell surface. The cells were washed with cold PBS and PE-labeled anti-human IgG Fc secondary antibody (Biolegend) was added and incubated for 30 min at 4 °C. The cells were collected for analysis.

## Antibody labeling and intracellular localization assays

AF03-NL and AF-03 were labeled with iQue Human Antibody Internalization Reagent (Sartorius) at the molar ratio of 1:3, respectively according to the manufacturer's instructions, protected from light, and incubated for 15 min at 37 °C. HEK293T cells expressing CI-MPR were added and incubated at 37 °C for 2 hr. Internalization of mAbs was detected by flow cytometry.

AF03-NL and AF-03 were covalently labeled with pH-sensitive pHrodo red succinimidyl ester (Thermo) according to the manufacturer's instructions. Antibodies were incubated with 10-fold molar excess of pHrodo red succinimidyl ester for 1 hr at room temperature. Excess unconjugated dye was removed using PD-10 desalting columns (GE Healthcare). pHrodo Red-labeled antibodies were exchanged into HEPES buffer and concentrated in an Amicon Ultra centrifugal filter unit with a nominal molecular weight cutoff of 30 kDa. Antibody concentration and degree of labeling was determined according to the manufacturer's instructions. HEK293T cells ($1~2 \times 10^5$ cells per dish) were cultured

overnight in a confocal dish pre-treated with polylysine (Beyotime) and then incubated with pHrodo Red-labeled AF-03 and AF03-NL (20 µg/ml) at 37 °C for 30 min. Unbound antibodies were removed by washing with cold PBS. As well, cells were stained with cell membrane dye (DiD) (Thermo) and Hoechst33342 (Thermo) at 37 °C for 15 min. Single cells were analyzed for pHrodo Red fluorescence on confocal microscopy (dragonfly 200).

### CI-MPR knockin and knockdown

Huh7 Cells were seeded in six-well plates at $3 \times 10^5$ cells/well and transfected with 2 µg CI-MPR expression plasmids (JetPRIME) when cells reached 40–60% confluence and cultured for 24 hr. For CI-MPR silencing, HEK293T cells were seeded in 6-well plates and transfected with siRNA- CI-MPR (Ribbio) and cultured for 48–72 hr. CI-MPR expression was detected by flow cytometry.

### Flow cytometry

The cells were harvested and stained with FITC-conjugated anti-CI-MPR antibody (Biolegend) on ice for 30 min. Cells were washed twice and detected on FACSAria II (BD Biosciences). Data analysis was performed using the FlowJo software.

### Statistical analyses

Data were analyzed, and the graphs were plotted using Prism software (GraphPad Prism 8, San Diego, USA). The data are presented as the mean ± standard error. Statistical differences were compared using unpaired t-tests or ANOVA. The $p < 0.05$ was considered statistically significant.

## Acknowledgements

This work is granted by the National Natural Science Foundation of China (81672803, 31771010, 81871252).

## Additional information

### Funding

| Funder | Grant reference number | Author |
| --- | --- | --- |
| National Natural Science Foundation of China | 81672803 | Guojiang Chen |
| National Natural Science Foundation of China | 31771010 | Yanchun Shi |
| National Natural Science Foundation of China | 81871252 | Yanchun Shi |

The funders had no role in study design, data collection and interpretation, or the decision to submit the work for publication.

### Author contributions

Yuting Zhang, Conceptualization, Data curation, Formal analysis, Writing - original draft; Min Zhang, Haiyan Wu, Xinwei Wang, Conceptualization, Data curation, Formal analysis; Hang Zheng, Formal analysis, Methodology; Junjuan Feng, Xinying Li, Methodology; Jing Wang, Longlong Luo, He Xiao, Chunxia Qiao, Yuanqiang Zheng, Investigation; Weijin Huang, Youchun Wang, Resources; Yi Wang, Yanchun Shi, Funding acquisition, Project administration; Jiannan Feng, Guojiang Chen, Funding acquisition, Writing - original draft, Project administration, Writing - review and editing

### Author ORCIDs

Jing Wang  http://orcid.org/0000-0001-9413-2378
Guojiang Chen  http://orcid.org/0009-0003-7289-0495

### Ethics

The animal study was reviewed and approved by the Institutional Animal Care and Use Committee of Academy of Military Medical Sciences(IACUC-DWZX-2020-697), which has been included in the materials and methods section.

Reviewer #2 (Public Review): https://doi.org/10.7554/eLife.91181.3.sa1
Author Response https://doi.org/10.7554/eLife.91181.3.sa2

## Additional files

### Supplementary files

• Supplementary file 1. Numerical data for *Figure 2C*.

• MDAR checklist

### Data availability

All data generated or analysed during this study are included in the manuscript and supporting files; Source data files have been provided.

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
