## [Editor Report · eLife assessment]

In this **valuable** study, the discovery and subsequent design of the AF03-NL chimeric antibody led to a tool for studying filoviruses and provides a possible blueprint for future therapeutics. In general, the data presented are **solid**, although further improvements can be made in the overall presentation of the results. The work will be of interest to virologists studying antibodies.

---

## [Referee Report · Reviewer #2 (Public Review)]

Summary:

The authors describe the discovery of a filovirus neutralizing antibody, AF03, by phage display, and its subsequent improvements to include NPC2 that resulted in greater breadth of neutralization. Overall, the manuscript is much improved from first review.

While the authors only use docking studies and this does not convincingly map the AF03 epitope, they do provide compelling evidence that residues Q128, N129, and possibly C226 are part of the epitope or at least close enough to affect binding and neutralisation. This is not conclusive support for their assumption that AF03 targets the NPC1 binding site. However, the authors do show that AF03 competes for MR78 binding to its epitope (in the NPC1 binding site), and this is enough to roughly place the epitope in this region (barring the possibility of an adjacent binding site with steric occlusion of the MR78 epitope).

The authors provide evidence for broad neutralisation, and also provide good support for the internalization of AF03-NL as the mechanism for improved breadth over the original AF03 antibody.

Strengths:

This study shows convincing binding to Marburgvirus GP and neutralization of Marburg viruses by AF03, as well as convincing neutralization of Ebolaviruses by AF03-NL. While there is not good separation of PE-stained populations by FACS in figure 5A, the cell staining data in Figure 5C are compelling to a non-expert in endosomal staining like myself. The control experiments in Figure 7 are compelling showing neutralization by AF03-NL but not AF03 or NPC2 alone or in combination. Altogether these data support the internalisation and stabilisation mechanism that is proposed for the gain in neutralization breadth observed for Ebolaviruses by AF03-NL over AF03 alone.

Weaknesses:

To support their affinity measurements, the authors argue that they show GP is a monomer in Figure 1A by SDS-PAGE. SDS-PAGE cannot be used to assess oligomerisation of GP. Native PAGE or size exclusion profiles would have been better suited to this purpose. If affinity was calculated on a 1GP:2IgG binding sites as the authors imply, then the affinity data are incorrect due to avidity effects. As suggested by a previous reviewer, using monomeric Fab would solve this problem.

The information for figure 2 states: "we investigated if this mutated MARV species was STILL sensitive o AF-03 treatment". But, "we sought to determine whether AF-03 could impede pseudotyped MARV viral entry" only happens in Figure 3. This information for figure 3 has now already been determined in Figure 2 where wildtype MARV is neutralised (black curves) introducing redundancy. The authors should first show that AF-03 can neutralise MARV pseudotyped virus, and then assess whether mutants are STILL sensitive to AF-03.

Figure 1: The visualisation of AF03 modelling and docking is better on a white background, but still difficult to interpret as currently presented. The labels of predicted contact residues are still impossible to read, and the yellow text does not show. As suggested previously, a zoom-in showing predicted co-location with Q128 and N129 would show these data better. It would also be useful to orient the reader with respect to trimeric membrane bound GP.

Figure 2: The presentation of these data is much improved and support the text.

Figure 3: The presentation of these data is much improved and support the text.

Figure 4: The presentation of these data is much improved and support the text.

Figure 5: The presentation of these data is much improved and support the text.

---

## [Author Response]

The following is the authors’ response to the original reviews.

**eLife assessment**
In this valuable study, the discovery and subsequent design of the AF03-NL chimeric antibody yielded a tool for studying filoviruses and provides a possible blueprint for future therapeutics. However, the data are incomplete and not presented clearly, which obscures flaws in the analyses and leaves unexplained phenomena. The work will be of interest to virologists studying antibodies.

Author response: Thank for your very valuable comments. The ms has been revised substantially and some new data have been added to further support the conclusions.

**Public Reviews:**

**Reviewer #1 (Public Review):**
Summary and Strengths:Zhang et al. conducted a study in which they isolated and characterized a Marburg virus (MARV) glycoprotein-specific antibody, AF-03. The antibody was obtained from a phage-display library. The study shows that AF-03 competes with the previously characterized MARV-neutralizing antibody MR78, which binds to the virus's receptor binding site. The authors also performed GP mutagenesis experiments to confirm that AF-03 binds near the receptor binding site. In addition, the study confirmed that AF-03, like MR78, can neutralize Ebola viruses with cleaved glycoproteins. Finally, the authors demonstrated that NPC2-fused AF-03 was effective in neutralizing several filovirus species.Weaknesses:(1) The main premise of this study is unclear. Flyak et al. in 2015 described the isolation and characterization of a large panel of neutralizing antibodies from a Marburg survivor (Flyak et al., Cell, 2015). Based on biochemical and structural characterization, Flyak proposed that the Marburg neutralizing antibodies bind to the NPC1 receptor binding side. In the same study, it has been shown that several MARV-neutralizing antibodies can bind to cleaved Ebola glycoproteins that were enzymatically treated to remove the mucin-like domain and glycan cap. In the following study, it has been shown that the bispecific-antibody strategy can be used to deliver Marburg-specific antibodies into the endosome, where they can neutralize Ebola viruses (Wec et al., Science 2016). Finally, the use of lysosome-resident protein NPC2 to deliver antibody cargos to late endosomes has been previously described (Wirchnianski et al., Front. Immunol, 2021). The above-mentioned studies are not referenced in the introduction. The authors state that "there is no licensed treatment or vaccine for Marburg [virus] infection." While this is true, there are human antibodies that recognize neutralizing epitopes - that information can't be excluded while providing the rationale for the study. Furthermore, the authors use the word "novel" to describe the AF-03 antibody. How novel is AF-03 if multiple Marburg-neutralizing antibodies were previously characterized in multiple studies? Since AF-03 competes with previously characterized MR78, it binds to the same antigenic region as MR78. AF-03 also has comparable neutralization potency as MR78.

Author response: Thank for your valuable advice. In terms of the novelty of AF-03, the inhibition assay indicates that Q128/N129/C226 functions as key amino acids responsible for AF-03 neutralization given that the neutralizing capacity of AF-03 to pesudotyped virus harboring these mutants is impaired (see revised Fig. 2A left panel). Furthermore, ELISA assays show that mutation of Q128S-N129S or C226Y significantly disrupts the binding of GP to AF-03, while the neutralizing and binding capacity of MR78 to mutant GP and pseudovirus harboring C226Y instead of Q128S-N129S is not almost affected (see revised Fig. 2A right panel and 2B). Considering the fact that AF-03 and MR78 could compete with each other to bind to MARV GP (Fig. 2D). we thus make a conclusion that the epitopes of these two mAbs overlapped partially. Therefore, AF-03 is not a clone of MR78 and is a novel neutralizing mAb to MARV.

The work from Wirchnianski and colleagues has been referenced actually in the ms (see Ref. 38). Although our strategy for the design of broad-spectrum neutralizing antibody refers to their work, we further expand the species being evaluated including RAVN and mutated EBOV strains. The results show that NPC2-fused AF-03 exhibits neutralizing activity to 10 filovirus species and 17 EBOV mutants (Fig. 6A and B). The work by Flyak et al. in 2015 that described the isolation and characterization of a large panel of neutralizing antibodies from a Marburg survivor has been cited in Introduction section accordingly.

(2) Without the AF-03-MARV GP crystal structure, it's unclear how van der Waals interactions, H-bonds, and polar and electrostatic interactions can be evaluated. While authors use computer-guided homology modeling, this technique can't be used to determine critical interactions. Furthermore, Flyak et al. reported that binding to the NPC1 receptor binding site is the main mechanism of Marburg virus neutralization by human monoclonal antibodies. Since both AF-03 (this study) and MR78 (Flyak study) competed with each other, that information alone was sufficient for GP mutagenesis experiments that identified the NPC1 receptor binding site as the main region for mutagenesis.

Author response: Computer-guided homology modeling has been exploited successfully in our lab to determine key residues responsible for the interaction between antigen and mAbs (Immunol Res. 2015, 62:377; Scand J Immunol. 2019, 90:e12777; Sci Rep. 2022, 12:8469; Front Immunol. 2022, 13:831536). We refer to the crystal structure of MARV GP and the complex of MR78 and GP reported previously (Cell 2015, 160:904) and then model the complex of MARV GP and AF-03. Although AF-03 and MR78 compete with each other, we show that the epitopes of these two mAbs just overlap partially (Fig. 2A-D).

(3) The AF-03-GP affinity measurements were performed using bivalent IgG molecules and trimeric GP molecules. This format does not allow accurate measurements of affinity due to the avidity effect. The reported KD value is abnormally low due to avidity effects. The authors need to repeat the affinity experiments by immobilizing trimeric GPs and then adding monovalent AF-03 Fab.

Author response: As shown in Fig. 1A, GP protein used in this work is not trimer but largely monomer composed of MLD-deleted GP1 and GP2, which may at a certain extent weaken the engagement between GP and AF-03. It is noteworthy that we re-done the SPR assays for the binding of AF-03 to GP and show that KD value is 4.71x10-11M (see revised Fig. 1C). This GP protein is thus available to the evaluation of mAb affinity. In addition, it is reasonable to utilize bivalent IgG to detect the affinity of mAb to monomeric GP since the affinity likely decreases significantly when monovalent Fab is used.

**Reviewer #2 (Public Review):**
Summary:The authors describe the discovery of a filovirus neutralizing antibody, AF03, by phage display, and its subsequent improvements to include NPC2 that resulted in a greater breadth of neutralization. Overall, the manuscript would benefit from considerable grammatical review, which would improve the communication of each point to the reader. The authors do not convincingly map the AF03 epitope, nor do they provide any strong support for their assumption that AF03 targets the NPC1 binding site. However, the authors do show that AF03 competes for MR78 binding to its epitope, and provides good support for the internalization of AF03-NL as the mechanism for improved breadth over the original AF03 antibody.Strengths:This study shows convincing binding to Marburgvirus GP and neutralization of Marburg viruses by AF03, as well as convincing neutralization of Ebolaviruses by AF03-NL. While there are no distinct populations of PE-stained cells shown by FACS in Figure 5A, the cell staining data in Figure 5C are compelling to a non-expert in endosomal staining like me. The control experiments in Figure 7 are compelling showing neutralization by AF03-NL but not AF03 or NPC2 alone or in combination. Altogether these data support the internalisation and stabilisation mechanism that is proposed for the gain in neutralization breadth observed for Ebolaviruses by AF03-NL over AF03 alone.Weaknesses:Overall, this reviewer is of the opinion that this paper is constructed haphazardly. For instance, the neutralization of mutant pseudoviruses is shown in Figure 2 before the concept of pseudovirus neutralization by AF03 is introduced in Figure 3. Similarly, the control experiments for AF03+NPC2 are described in Figure 7 after the data for breadth of neutralization are shown in Figure 6. GP quality controls are shown in Figure 2 after GP ELISAs / BLI experiments are done in Figure 1. This is disorienting for the reader.

Author response: AF-03 production and its binding capacity to GP is determined in Fig. 1. The epitopes of AF-03 is identified in Fig. 2. The neutralizing activity of AF-03 to pseudotyped MARV in vitro and in vivo is detected in Fig. 3. The neutralizing activity of AF-03 to pseudotyped ebolavirus harboring cleaved GP is detected in Fig. 4. The endosome-delivering ability of AF03-NL is examined in Fig. 5. The neutralization of filovirus species and EBOV mutants by AF03-NL is detected in Fig. 6. The requirement of CI-MPR for neutralization activity of AF03-NL is determined in Fig. 7. We think that this arrangement is suitable.

Figure 1: The visualisation of AF03 modelling and docking endeavours is extremely difficult to interpret. Firstly, there is no effort to orient the non-specialist reader with respect to the Marburgvirus GP model. Secondly, from the figures presented it is impossible to tell if the Fv docks perfectly onto the GP surface, or if there are violent clashes between the deeply penetrating AF03 CDRs and GP. This information would be better presented on a white background, perhaps showing GP in surface view from multiple angles and slices. The authors attempt to label potential interactions, but these are impossible to read, and labels should be added separately to appropriately oriented zoomed-in views.

Author response: To be readily understood the rationale of computer-guided modeling, the descriptions in the Methods and Results section have been refined accordingly. In addition, the information of the theoretical structure was presented on white background (see revised Fig. 1D-F).

Figure 2: The neutralization of mutant pseudoviruses cannot be properly assessed using bar graphs. These data should be plotted as neutralization curves as they were done for the wild-type neutralization data in Figure 3. The authors conclude that Q128 & N129 are contact residues, but the neutralization data for this mutant appear odd as the lowest two concentrations of AF03 show higher neutralization than the second highest AF03 concentration. Neutralization of T204/Q205/T206 (green), Y218 (orange), K222 (blue), or C226 (purple) appears to be better than neutralization of the wild-type MARV. The authors do not discuss this oddity. What are the IC50's? The omission of antibody concentrations on the x-axis and missing IC50 values give a sense of obscuring the data, and the manuscript would benefit from greater transparency, and be much easier to interpret if these were included. I am intrigued that the Q128S/N129S mutant is reported as having little effect on the neutralization of MR78. The bar graph appears to show some effect (difficult to interpret without neutralization curves and IC50 data), and indeed PDB:5UQY seems to suggest that these amino acids form a central component of the MR78 epitope (Q128 forms potential hydrogen bonds with CDRH1 Y35 and CDRL3 Y91, while N129 packs against the MR78 CDRH3 and potentially makes additional polar contact with the backbone). Lastly, since neutralization was tested in both HEK293T cells and Huh7 cells in Figure 3, the authors should clarify which cells were used for neutralization in Figure 2.

Author response: Thank for your advice. Accordingly, in the revised ms, the neutralization curve of AF-03 and MR78 is presented in revised Fig. 2A. The neutralization of AF-03 to pseudotyped MARV harboring Q128S/N129S or C226Y is impaired significantly compared with WT MARV and those bearing other indicated mutations, while Q128S/N129S instead of C226Y mutation affect the neutralizing capacity of MR78 at a certain extent. This is consistent with the data on the binding of AF-03 or MR78 to MARV GP protein assayed by ELISA (see revised Fig. 2B). Overall, these results show that Q128/N129/C226 functions as key amino acids responsible for AF-03 neutralization.

Figure 3: The first two images in Figure 3C showing bioluminescent intensity from pseudovirus-injected mice pretreated with either 10mg/kg or 3mg/kg AF03 are identical images. This is apparent from the location, shape, and intensity of the bioluminescence, as well as the identical foot placement of each mouse in these two panels. Currently, this figure is incomplete and should be corrected to show the different mice treated with either 10mg/kg or 3mg/kg of AF03.

Author response: Thank for your carefulness. Indeed, it is our mistake. In the revised ms, this fault has been corrected. The correct images have been added (see revised Fig. 3C).

Figure 4 would benefit from a control experiment without antibodies comparing infection with GP-cleaved and GP-uncleaved pseudoviruses. The paragraph describing these data was also difficult to read and would benefit from additional grammatical review.

Author response: Accordingly, a control experiment comparing the infection of GP-cleaved with GP-uncleaved pseudoviruses is performed. The results show that The infection of pseudotyped ebolavirus harboring cleaved GP to host cells is comparable or stronger than those containing intact GP（see revised Fig. s1）. Therefore, the data in Fig. 4 support the inhibition of cell entry of ebolavirus species harboring cleaved GP by AF-03, which is not attributed to the possible impairment of cell entry capacity of GPcl-containing ebolavirus. In addition, the sentences have been modified to be read smoothly.

Figure 5: The authors should clarify in the methods section that the "mock" experiment included the PE anti-human IgG Fc antibody. Without this clarification, the lack of a distinct negative population in the FACS data could be interpreted as non-specific staining with PE. If the PE antibody was added at an equivalent concentration to all panels, what does the directionality of the arrowheads in Figure 5A (labelled PE) and 5B (labelled pHrodo Red) indicate?

Author response: Thank for your advice. In the revised version, we denote that the mock is actually a human IgG isotype in the figure legend. The arrowheads denote the fluorescence intensity of PE or pHrodo on the lateral axis of the plots. Of course, herein the percentage of PE or pHrodo-positive cells is shown.

Figure 6B: These data would benefit from the inclusion of IC50, transparency of antibody concentrations used, and consistency in the direction of antibody concentrations (increasing to the right or left of the x-axis) when compared to Figure 2.

Author response: The concentration of antibody titrated is shown in figure legends. The direction of antibody concentrations is unified throughout the paper. Although IC50 is not included, these data clearly show that AF03-NL rather than AF-03 prominently inhibits the cell entry of EBOV mutants.

**Reviewer #1 (Recommendations For The Authors):**
Line 143: anti-human should be anti-human.Line 223: From the SDS-PAGE results, it's not clear that the AF-03 was expressed in the eukaryotic cell line. Please, rephrase the sentence.Line 263: ELISA experiments can't be used to determine affinity.Line 394: Flyak et al. generated human antibodies from PBMC samples of Marburg survivors, not plasma samples.

Author response: According to reviewer's advice, the sentences have been modified or corrected to more accurately describe the results. As well, the grammatic errors in the ms have been corrected carefully.